# Minimising efficiency roll-off in high-brightness perovskite light-emitting diodes

Wei Zou[1], Renzhi Li[1], Shuting Zhang[1], Yunlong Liu[1,2], Nana Wang [1], Yu Cao[1], Yanfeng Miao[1], Mengmeng Xu[1], Qiang Guo[1], Dawei Di [3], Li Zhang[1], Chang Yi[1], Feng Gao[4], Richard H. Friend [3], Jianpu Wang [1] & Wei Huang[1,5,6]

Efficiency roll-off is a major issue for most types of light-emitting diodes (LEDs), and its origins remain controversial. Here we present investigations of the efficiency roll-off in perovskite LEDs based on two-dimensional layered perovskites. By simultaneously measuring electroluminescence and photoluminescence on a working device, supported by transient photoluminescence decay measurements, we conclude that the efficiency roll-off in perovskite LEDs is mainly due to luminescence quenching which is likely caused by non-radiative Auger recombination. This detrimental effect can be suppressed by increasing the width of quantum wells, which can be easily realized in the layered perovskites by tuning the ratio of large and small organic cations in the precursor solution. This approach leads to the realization of a perovskite LED with a record external quantum efficiency of 12.7%, and the efficiency remains to be high, at approximately 10%, under a high current density of 500 mA cm$^{-2}$.

[1] Key Laboratory of Flexible Electronics (KLOFE) & Institute of Advanced Materials (IAM), Jiangsu National Synergetic Innovation Center for Advanced Materials (SICAM), Nanjing Tech University (NanjingTech), 30 South Puzhu Road, 211816 Nanjing, China. [2] School of Physical Science and Information Technology, Liaocheng University, Liaocheng, 252059 Shandong, China. [3] Cavendish Laboratory, Cambridge University, JJ Thomson Avenue, Cambridge, CB3 0HE, UK. [4] Biomolecular and Organic Electronics, IFM, Linköping University, 58183 Linköping, Sweden. [5] Key Laboratory for Organic Electronics and Information Displays & Institute of Advanced Materials (IAM), Jiangsu National Synergetic Innovation Center for Advanced Materials (SICAM), Nanjing University of Posts & Telecommunications, 9 Wenyuan Road, 210023 Nanjing, China. [6] Shaanxi Institute of Flexible Electronics (SIFE), Northwestern Polytechnical University (NPU), 127 West Youyi Road, Xi'an, 710072 Shaanxi, China. Wei Zou and Renzhi Li contributed equally to this work. Correspondence and requests for materials should be addressed to J.W. (email: iamjpwang@njtech.edu.cn) or to W.H. (email: iamwhuang@nwpu.edu.cn)

L ow-temperature solution-processed metal halide perovskites show excellent luminescence and charge transport properties, high color purity, and tunable bandgap, making them promising for low-cost and high-performance light-emitting diode (LED) applications[1–4]. Compared to other similar technologies, such as organic LEDs and quantum-dot LEDs, perovskite LEDs can potentially achieve better efficiency at high brightness since its charge mobility is higher and luminescence quenching due to high exciton concentration is less significant[2,5]. Previous investigations of perovskite light-emitting diodes (PeLEDs) mainly focused on three-dimensional (3D) perovskites[1,2,6]. However, these 3D perovskites can suffer from large leakage current due to incomplete surface coverage when the film thickness is decreased to tens of nanometers in LEDs[1,2]. In order to obtain emissive perovskite thin films with good uniformity, PeLEDs based on perovskite multiple quantum wells (MQWs) have been recently developed[3,4,7]. Here, the term MQW refers to an assembly of different layered perovskites fabricated through solution processing. They are analogous to 'quantum wells' (QWs) by considering their energetic structures. They consist of inorganic potential wells and organic potential barriers. The bandgap of each QW is mainly determined by the number of PbI$_4$ monolayer sheets ($n$) within a layered perovskite. We note that the layered perovskite structure with a well-defined $n$ number can be referred to Ruddlesden-Popper phase[8,9]. High external quantum efficiency (EQE) up to 11.7% can be obtained with the MQW perovskites based LED because the lower energy gap QWs which generate electroluminescence (EL) are confined by higher bandgap QWs[4].

The application of QW structures is a key to obtain high-performance LEDs made with III–V semiconductors. However, it requires precise control of lattice parameter matching to avoid luminescence quenching caused by electronic defects[10]. In contrast, perovskite MQWs are self-organized nanostructures from low-temperature solution processing, showing excellent emission properties, with photoluminescence quantum efficiencies (PLQEs) of up to 67%[4]. Similar to the case of III-V LEDs, these perovskite MQWs facilitate confinement of charge carriers, enhancing the probability of radiative recombination (Fig. 1a). In addition, by employing perovskite MQWs, device operational stability is significantly enhanced[4], making reliable assessment of device physics possible for these LEDs.

For practical application of LEDs, a key indicator of device quality is its efficiency at high current densities (or brightness). For most LEDs, their efficiencies reduce at high current densities.

This behavior is termed as efficiency roll-off, which can be caused by either luminescence quenching (a reduction in luminescence efficiency due to non-radiative processes) or an excessive population of charge carriers passing through the device without forming electron-hole pairs[11–17]. Here, we show that typical PeLEDs based on MQWs also suffer from serious efficiency roll-off, with the EQE peaking at 30 mA cm$^{-2}$ and rolling-off by 55% (relative to the peak value) at 400 mA cm$^{-2}$ (Fig. 1b). We investigate the origin of the efficiency roll-off, by simultaneously measuring EL and photoluminescence (PL) on a working LED device. We find that the efficiency roll-off in PeLEDs is mainly due to luminescence quenching most likely caused by non-radiative Auger recombination, and the luminescence quenching effect can be suppressed by increasing the width of perovskite QWs. Based on our improved understanding of device operation, we are able to raise the efficiency of PeLEDs to 12.7% with significantly reduced roll-off.

## Results

**Origin of efficiency roll-off in PeLEDs.** The EL EQE roll-off at high intensities could be caused by either luminescence quenching or an excessive population of charge carriers passing through the device without forming electron-hole pairs[11–13,17]. In order to investigate which mechanism dominates the efficiency roll-off in the MQW perovskite LEDs, we measure the PLQE and EQE of the devices simultaneously by using low-frequency chopped illumination at very low intensity (0.03 mW cm$^{-2}$), when the LED device is in operation. Detailed experimental setup is presented in Supplementary Fig. 1. The LED device is based on NFPI$_7$ MQW perovskite film which is deposited from a precursor solution which contains 1-naphthylmethylamine iodide (NMAI), formamidinium iodide (FAI), and PbI$_2$ with a molar ratio of 2:1:2[4]. We find that the PL response decreases at high current densities, following the same trend as the EL EQE (Fig. 1b). The excellent correlation between the PLQE and EQE at high current intensities indicates that luminescence quenching is responsible for the EQE roll-off.

The quenched PL intensities (and hence the efficiency roll-off) at high intensities could result from either high carrier (and/or exciton) density or high electrical field across the emissive layer[18–20]. In order to disentangle these two different mechanisms, we monitor the PL response at negative bias, where the electric field is high and the carrier/exciton density is low. Figure 1b also shows that the PL decreases when the applied voltage is swept from 0.75 to −2 V (from below turn-on to reverse

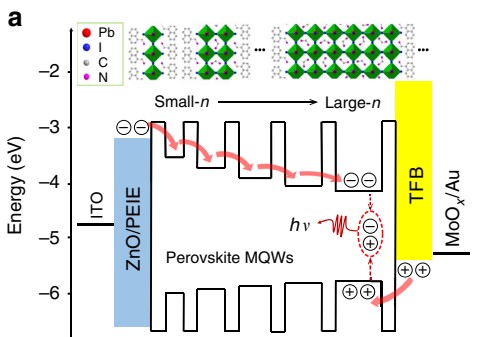
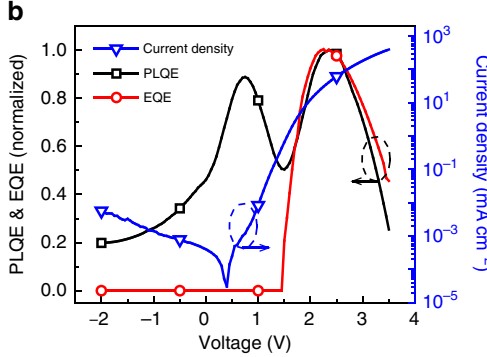

**Fig. 1** Device structure and efficiency roll-off of perovskite MQW LEDs. **a** Schematic representation of the flat-band energy level diagram and structures of the 30-nm thick perovskite MQW film which is an assembly of different layered lead halide perovskites with various $n$ numbers. The $n$ number determines the bandgap of each quantum well. The MQW structure enhances the probability of radiative recombination. **b** Dependence of current density (blue triangles), normalized PLQE (black square), and EQE (red circle) on the driving voltage. The PLQE and EQE were measured simultaneously on a working LED device. The excellent correlation between the PLQE and EQE at high current intensities indicates that luminescence quenching is responsible for the EQE roll-off

bias). The PL intensity also decreases when the bias is swept from 0.75 to 1.5 V (from below turn-on to near turn-on), when the current density is still very small. We consider that the internal electrical field across the perovskite film is minimized (approaching zero) under a forward bias of 0.75 V, corresponding to a flat-band condition for the emissive layer, leading to the minimum luminescence quenching. When the bias is swept from 0.75 to 1.5 V, the PLQE drops due to the increasing electrical field across the perovskite layer. We are aware that the electrical field distribution across different regions of the device can be complex, considering band bending due to interfacial effects and dielectric constant variations in the multilayer structure. These results indicate that at low carrier densities produced by photoexcitation, the internal electrical field in the device quenches the luminescence of MQW perovskites.

For QWs, the electrical field-induced PL quenching can result from either quantum confined Stark effect (QCSE)[11,21] or charge separation[20,22–24]. Here, we can rule out the former, because the emission peaks blue-shift with an increasing electrical field (Fig. 2a), in contrast to the characteristic red-shift associated with QCSE[21]. We suggest the PL quenching in our case is caused by field-induced charge separation. The blue-shifted emission spectra under electric field is consistent with the scenario that excitons from QWs of different widths have different binding energies[4], and excitons with large binding energies (resulting from narrow well width) are relatively less affected by the field. Indeed, this can be supported by measuring the spectral dependence of the quenching, $\phi(E) = \frac{I(0)-I(E)}{I(0)}$. As shown in Supplementary Fig. 2, the higher energy emission from QWs of narrow width experiences much weaker quenching under the same electrical field. Moreover, the $J$–$V$ curves of device under various illumination intensities (Fig. 2b) can clearly show that the charges separate under negative bias.

In order to gain more insights into the field-induced charge separation process, we have also performed PL decay measurements under different biases. As shown in Fig. 2c, the lifetime decreases with an increasing electrical field, indicating that the charge separation is likely followed by a charge tunneling process which removes the charges from the QWs. With only exciton separation, we would expect a longer lifetime under higher field, since the separated charges are still confined within the well and can eventually recombine with a probability of emission[25]. Indeed, the process of Fowler–Nordheim tunneling of charges after exciton separation has been observed in inorganic QWs[25,26].

So far, we have shown that the luminescence of our MQW perovskites is quenched by electrical field through a charge separation process. However, Fig. 1b also shows that the PL starts to increase after the bias exceeds the turn-on voltage (1.5 V, when the device emits light) for bipolar charge injection and EL. We note that the observation of a slightly lower turn-on voltage may be possible with more sensitive experimental setups. Under these conditions, where the density of photogenerated carriers is much lower than the density of electrically injected charges, we expect the recombination of the photogenerated charges occurs predominantly with the injected charges, so that the PL response now tracks the EL efficiency, as is evident in Fig. 1b.

We note the EL emission spectra under a high forward bias is almost identical to the PL spectrum at near-zero internal field (as explained previously, this is when the bias is 0.75 V) (Fig. 2a), suggesting the importance of screening effect during LED device operation under high bias. We also notice that the peak PLQE (at 2.4 V) even exceeds the local maximum at 0.75 V, where there is no internal field. This can be explained by the trap-filling effects: injected charge carriers fill in the trap states, eliminating trap-mediated non-radiative recombination and resulting in enhanced PL intensity relative to the zero charge injection case[5].

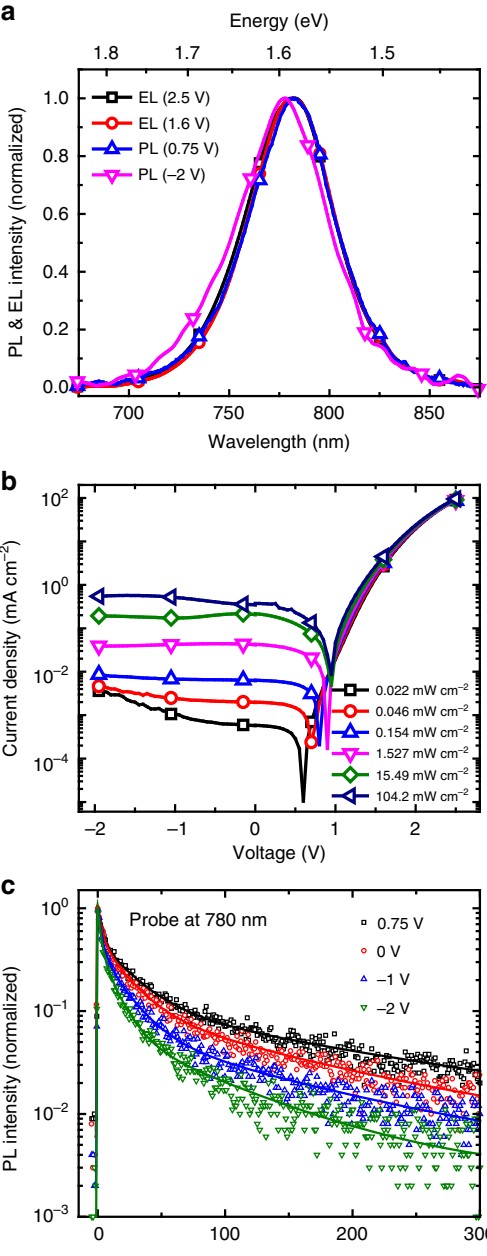

**Fig. 2** Electrical field-dependent characteristics of perovskite MQW LEDs. **a** Normalized EL and PL spectra of the device under various bias voltages. EL (2.5 V, black square, 1.6 V, red circle), PL (0.75 V, blue triangle, −2 V, magenta triangle). **b** Dependence of current density on the driving voltage under various illumination intensities. **c** Time-resolved PL decay transients of perovskite films under various bias voltages, measured at 780 nm

In order to provide evidence for our argument on the screening effect in the perovskite MQW LED under high bias, we measure the bias-dependent PL intensities under various photoexcitation intensities. In this way, we vary the density of photogenerated carriers, analogously to the electrically injected carriers in LED device operation. As shown in Supplementary Fig. 3, with increasing excitation intensities, the effect of PL quenching decreases and the trough due to the competition between field-induced quenching effect and screening effect gradually disappears. Therefore, this measurement further confirms that electrical field-induced quenching becomes less important under high excitation intensities.

Based on the above analysis, we can exclude that the electrical field-induced quenching is the main reason for the EQE drop at high current densities. Then we proceed to investigate the most critical regime of the PLQE/EQE-voltage curves, i.e., the emission drop (for both EL and PL) under high current densities (Fig. 1b). We can first rule out the effects of heat-induced luminescence quenching by measuring the PLQE under pulsed laser excitation (Fig. 3a), which also shows similar efficiency drop. The PLQE starts to decrease at excitation fluences higher than 30 nJ cm$^{-2}$ per pulse (corresponding to a carrier density of $3 \times 10^{15}$ cm$^{-3}$), below which the PLQE is relatively constant. More importantly, transient measurements show that the PL lifetime is almost constant when the PLQE is constant, while it decreases significantly as the PLQE drops (Fig. 3b). Detailed fitting results of PL lifetime change under various excitation intensities are presented in Supplementary Fig. 4. Steady-state measurements of PLQE show consistent results (Fig. 3c). The PLQE and PL lifetime simultaneously decrease at high excitation intensities, suggesting that the efficiency roll-off is likely due to an Auger recombination[27–29]. We notice that here the Auger process kicks in at relatively low carrier densities compared to 3D perovskites, for which it becomes dominant at a carrier density of $1 \times 10^{18}$ cm$^{-3}$ [28]. The reason can be that in our MQW structures, the charge carriers are concentrated in the QWs with lower bandgaps which only occupy a small portion of the MQW film[4], so that the local carrier density in these QWs is much higher than the 3D case with the same total carrier density.

**High-performance LEDs with reduced efficiency roll-off**. If the efficiency roll-off in our MQW perovskite LEDs is caused by an Auger recombination, then it can be improved by reducing the local carrier density in QWs. A feasible approach is to increase the width of lower-bandgap QWs to reduce the local carrier density (Fig. 3d). We note that in GaN LEDs, precisely-controlled double heterostructure (DH) has been fabricated to suppress Auger recombination[30]. Compared to GaN-based MQW LEDs,

the peak EQEs of these DH LEDs is lower, likely due to the difficulty of avoiding electronic defects in these double heterostructures. While in perovskite MQW devices, without scarifying the luminescence properties, we can easily tune the well width by controlling the ratio of large and small organic cations in the precursor solution. As we increase the concentration of the small organic cations (FA) by changing the molar ratio of NMAI, FAI, and PbI$_2$ from 2:1:2 to 2:1.9:2, the main PL peak of the MQW film exhibits a 5.6-nm red-shift (Fig. 3e), indicating the formation of wider QWs[3,4]. The X-ray diffraction (XRD) measurement shows that the 2:1.9:2 perovskite MQW film contains more large-$n$ QWs with large grain size (Supplementary Fig. 5)[31,32] which is consistent with the scanning electron microscopy (SEM) image (Supplementary Fig. 6), further confirming the formation of wider QWs.

To confirm whether the wider QWs can reduce luminescence efficiency roll-off at high excitations, we have also performed fluence-dependent PLQE measurements under pulsed and continuous-wave (cw) laser excitations. As expected, by distributing the carriers in wider QWs, the PLQE remains constant till an excitation fluence of up to 80 nJ cm$^{-2}$ per pulse (in contrast to 30 nJ cm$^{-2}$ per pulse in the narrower QWs) (Fig. 3f). The maximum PLQE from the steady-state measurements is closed to 8 mW cm$^{-2}$ (in contrast to 3 mW cm$^{-2}$ in the narrow-width case), with a steadier fall-off as the excitation increases (Fig. 3c). The PL lifetimes under high excitations are longer for the wider QWs compared to that for the narrower wells (Fig. 3b, g), and detailed fitting results are shown in Supplementary Fig. 4. Therefore, these optical measurements suggest that the Auger recombination is likely suppressed in 2:1.9:2 perovskite MQW films with wider QWs.

Based on these improved perovskite MQWs, we find that the EL device performance is clearly enhanced (Fig. 4). Figure 4a shows that under high current densities, the EL intensity of the 2:1.9:2 device is significantly enhanced compared to the 2:1:2 device. The EQE of the 2:1.9:2 device peaks at 80 mA cm$^{-2}$ (2.3

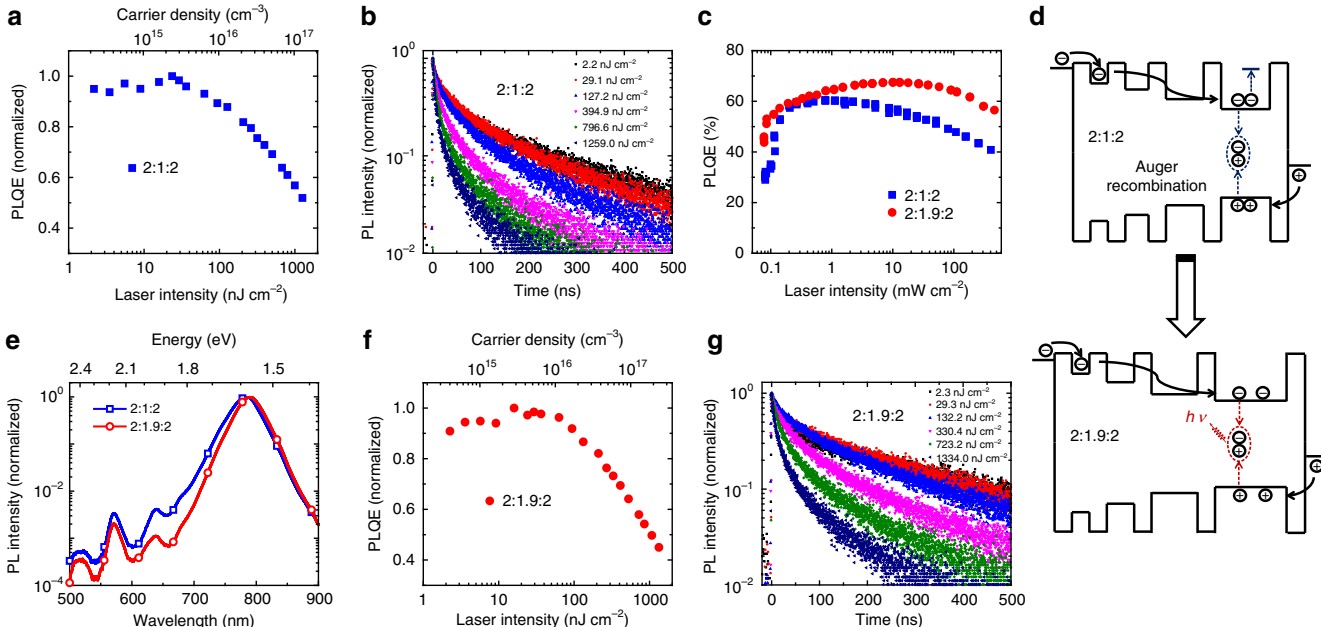

**Fig. 3** Fluence-dependent emission properties of perovskite MQW films. **a** Excitation-intensity-dependent PLQE of the 2:1:2 NFPI$_7$ MQW film under pulsed laser excitation. **b** Time-resolved PL decay transients of the 2:1:2 NFPI$_7$ MQW film under different excitation intensities. **c** Excitation-intensity-dependent PLQEs of the 2:1:2 and 2:1.9:2 NFPI$_7$ MQW films under continuous-wave laser excitation. **d** Schematic representation of charge recombination in MQWs with different well widths. **e** PL (445 nm excitation) spectra of the 2:1:2 and 2:1.9:2 NFPI$_7$ MQW films. **f** Excitation-intensity-dependent PLQE of the 2:1.9:2 NFPI$_7$ MQW film under pulsed laser excitation. **g** Time-resolved PL decay transients of the 2:1.9:2 NFPI$_7$ MQW film under different excitation intensities

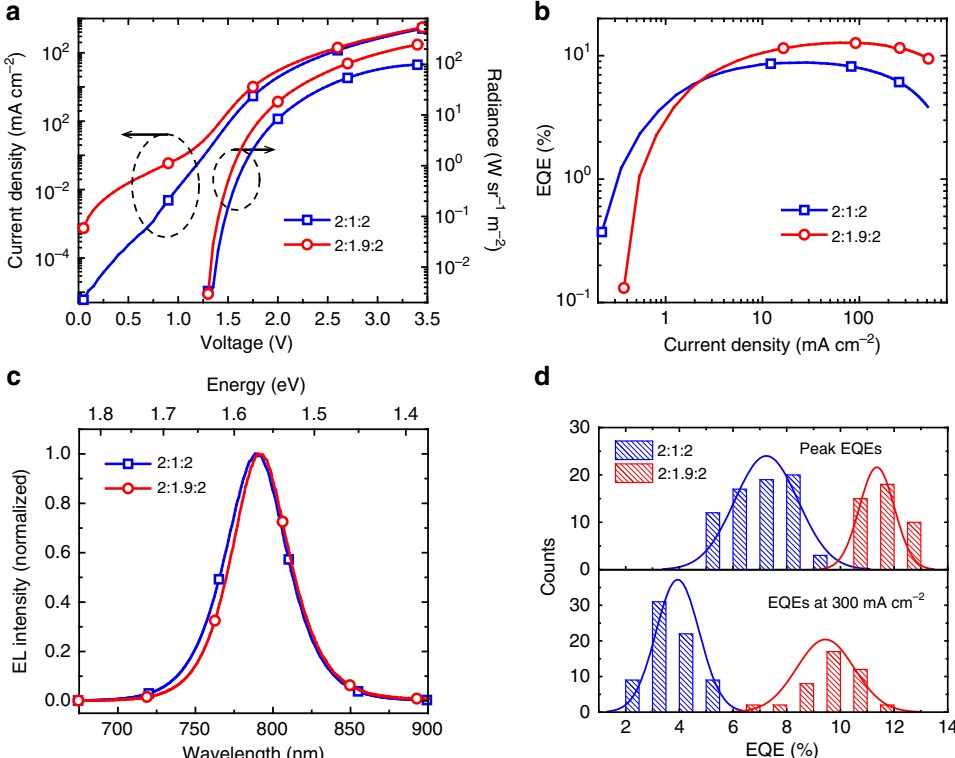

**Fig. 4** Optoelectronic characteristics of the perovskite MQW LEDs. **a** Dependence of current density and radiance on the driving voltage. The peak radiance of 254 W sr$^{-1}$ m$^{-2}$ is obtained under 3.45 V for the 2:1.9:2 devices. This is a brightness record for solution-processed NIR LEDs. **b** EQE versus current density. For the 2:1.9:2 MQW LED, a peak EQE of 12.7% is achieved at a current density of 80 mA cm$^{-2}$. The EQE of the 2:1.9:2 device remains to be approximately 10% at 500 mA cm$^{-2}$, due to a significantly suppressed EQE roll-off. **c** EL spectra of the 2:1:2 and 2:1.9:2 MQW LED devices. **d** Histograms of peak EQEs and EQEs at a high current density (300 mA cm$^{-2}$). Statistics of 71 2:1:2 devices show an average peak EQE of 7.2% (a relative standard deviation of 16.3%) and an average EQE of 4.0% at 300 mA cm$^{-2}$ (a relative standard deviation of 20%). Peak EQEs of 43 2:1.9:2 devices were measured under the same condition, which shows an average EQE of 11.3% (a relative standard deviation of 5.7%) and an average EQE of 9.4% at 300 mA cm$^{-2}$ (a relative standard deviation of 12.6%)

V), reaching a high value of 12.7%, while the EQE of the 2:1:2 device peaks at 30 mA cm$^{-2}$ (2.1 V) with a value of 8.7% (Fig. 4b). Moreover, the efficiency roll-off with the 2:1.9:2 device is significantly suppressed, and the EQE remains high, at approximately 10%, under a high current density of 500 mA cm$^{-2}$. Compared to the 2:1:2 device, the EL emission spectrum of the 2:1.9:2 device is slightly red-shifted (Fig. 4c), which is consistent with the emission from the wider QWs. The EL measurement is in good agreement with the PL measurement, also suggesting the device performance improvement is likely a result of reduced Auger recombination. We note that the 12.7% EQE is a new efficiency record for PeLEDs[4]. In addition, due to the slow roll-off, the device shows a high radiance of 254 W sr$^{-1}$ m$^{-2}$ (at 3.45 V), which is also a record for solution-processed near-infrared (NIR) LEDs[4]. As shown in Fig. 4d, the EQE histogram for 43 2:1.9:2 devices shows a remarkably high average peak EQE of 11.3% with a small relative standard deviation of 5.7%, compared to those from 71 2:1:2 devices measured under the same condition (a 7.2% average peak EQE and a 16.3% relative standard deviation). Moreover, the average EQE at high current density (300 mA cm$^{-2}$) is 9.4% for these 2:1.9:2 devices, which is more than twice of the value of 2:1:2 devices (4.0%) at the same current density. The device lifetime is dependent on the current densities, as shown in Supplementary Fig. 7. The lifetime ($T_{50}$, time to half of the initial brightness) of the 2:1.9:2 device (with a simple glass–epoxy encapsulation) under a constant current density of 100 mA cm$^{-2}$ is approximately 30 min.

## Discussion

By simultaneously measuring the EL and PL on a working device, supported by transient PL decay measurements, we conclude that the efficiency roll-off in perovskite LEDs is mainly due to luminescence quenching which is most likely caused by an Auger recombination. In GaN-based LEDs, they usually require sophisticated control to reduce the luminescence quenching[30]. While in the solution-processed MQW perovskite LEDs, the processing simplicity of the MQW structures allows easy tunability of the QW width, resulting in reduced luminescence quenching and enhanced device performance. Our work provides a guide to achieve high-efficiency, high-brightness, and low-roll-off perovskite LEDs, which may also have implications for the development of other kinds of LEDs.

## Methods

**Preparation of perovskite MQW.** Here, the 2:1:2 and 2:1.9:2 NFPI$_7$ perovskite layers were deposited from spin-coating of 7 wt% precursor solutions of NMAI, FAI, and PbI$_2$ with molar ratios of 2:1:2 and 2:1.9:2 dissolved in a solution of $N,N$-dimethylformamide (DMF), respectively. The details of materials preparation can be found in the previous report[4].

**Device fabrication.** The devices were fabricated on a glass substrate coated with indium tin oxide and has the structure: ITO/polyethylenimine ethoxylated (PEIE)-modified ZnO (20 nm)/perovskite (30 nm)/poly(9,9-dioctyl-fluorene-co-N-(4-butylphenyl)diphenylamine) (TFB, 40 nm)/MoO$_3$ (7 nm)/Au (60 nm). ZnO, PEIE, perovskite, and TFB were fabricated using spin-coating processes. MoO$_3$ and Au were thermally evaporated. The active area of the device is 1 mm × 3 mm, which is the overlapping area of the ITO bottom electrode and the Au top electrode.

**Device characterization**. The EL devices were characterized on top of the integration sphere, and notably, only forward light emission could be collected. The details of fabrication and characterization can be found in the previous report[4].

**XRD measurement**. XRD spectra of the perovskite films were obtained by a Philips X-ray diffractometer with CuKα radiation.

**SEM measurement**. The morphology of the perovskite films was recorded by using a JEOL JSM-7800F SEM.

**Steady-state PL measurement**. Steady-state PL spectra of the perovskite films were recorded by using a fluorescent spectrophotometer (F-4600, HITACHI) with a 200 W Xe lamp as an excitation source.

**Time correlated single photon counting measurement**. Transient PL spectra and the PL decay were obtained by using an Edinburgh Instruments spectrometer (FLS980) with a 445-nm pulsed laser (less than 100 ps, 1 MHz, YSL Supercontinuum Source SC-PRO). The total instrument response function (IRF) was less than 130 ps, and temporal resolution was less than 20 ps. The PL curves can be well fitted by the thermalized stretching exponential line shape,

$$I(t) = I_1 \exp\left(-\frac{t}{\tau_1}\right) + I_2 \exp\left[-\left(\frac{t}{\tau_2}\right)^{\beta}\right] \quad (1)$$

where $\tau_i$ is the decay time and $\beta$ is the decay exponent[33]. $\tau_1$ is fast process lifetime related to Auger recombination or trap-assisted charge recombination, and $\tau_2$ can be described as bi-molecular recombination and other long lifetime processes[33]. Supplementary Fig. 4 presents the fitting results. At low excitations, it shows $\tau_1$ slightly increases when the excitation increases, indicating the importance of trap-assisted charge recombination. When the excitation becomes high and the PLQE decreases, $\tau_1$ starts to decrease, which is consistent with Auger recombination-induced luminescence quenching effect. $\tau_2$ increases with an increasing light intensity, consistent with an enhanced bi-molecular recombination with an increased carrier density.

**PLQE measurement**. For the PLQE measurement, a three-step technique was used through the combination of a 445-nm CW laser, an optical fiber, a spectrometer, and an integrating sphere[34].

**Simultaneous measurements of PL and EL**. In order to identify the dominant process in efficiency roll-off, PL and EL were measured simultaneously. A lock-in amplifier was used to measure the PL signal, which was kept far lower than the EL intensity (PL/EL less than 0.001% at 2 V)[14,35]. Supplementary Fig. 1 shows the schematic diagram of the measurement setup. The LED devices were excited by a cw 445 nm laser with an optical chopper for frequency modulation (930 Hz). The excitation light was focused onto a circular spot with a diameter of 0.4 mm, so that the spot size is much smaller than the device size. A Keithley 2400 source-meter was used as the voltage source of the LED device. The scanning speed is 0.05 V s⁻¹ and the dwell time is 1 s for each voltage step. The EL and PL were obtained by the combination of a lock-in amplifier (SR830), an electric-meter (Keithley 2000), and a photodector (Thorlabs PDA100A). A beam splitter and another PDA100A photodetector were used to monitor the excitation light intensity.

**Photo-generated carrier density calculation**. Photo-generation carrier density can be calculated as:

$$\rho_{\text{photocarrier}} = \frac{\text{light fluence density of single pulse} \times \text{light absorbance}}{\text{photo energy} \times \text{film thickness}}. \quad (2)$$

Here, the film thickness is 30 nm. Light absorbance is converted from ultraviolet–visible absorbance spectra, which were recorded by an ultraviolet–visible spectrophotometer with an integrating sphere (Cary 5000, Agilent).

**Data availability**. The data that support the finding of this study are available from the corresponding author upon reasonable request.

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

## Acknowledgements

This work is financially supported by the National Basic Research Program of China-Fundamental Studies of Perovskite Solar Cells (2015CB932200), the Joint Research Program between China and European Union (2016YFE0112000), the Major Research Plan of the National Natural Science Foundation of China (91733302), the Natural Science Foundation of Jiangsu Province, China (BK20150043), the National Natural Science Foundation of China (11474164, 61405091, 61634001), the National Science Fund for Distinguished Young Scholars (61725502), the Synergetic Innovation Center for Organic Electronics and Information Displays, the Swedish Government Strategic Research Area in Materials Science on Functional Materials at Linköping University (Faculty Grant SFO-Mat-LiU # 2009-00971), the Swedish Research Council (VR), and the European Commission Marie Skłodowska-Curie actions (691210).

## Author contributions

J.W. conceived the project. J.W. and W.H. supervised the work. W.Z. and R.L. managed to set up the optical testing systems and took the measurements with the assistance of Y.L. and M.X. S.Z. carried out the device fabrication and characterizations with the assistance of N.W., L.Z., and Q.G. C.Y. and Y.C. synthesized the NMAI. Y.M. took part in TCSPC measurement. J.W., W.Z., N.W., F.G., D.D., and R.H.F. analyzed the data. J.W. and F.G. wrote the first draft of the manuscript. D.D., R.H.F., and W.H. provided major revisions. All authors commented on the manuscript.

## Additional information

**Competing interests:** The authors declare no competing financial interests.

