## [Peer Review File · Nature Communications]

Reviewers' comments:

Reviewer #1 (Remarks to the Author):

In this paper, the authors fabricate perovskite LEDs with low efficiency roll-off at high injection current density, and attempt to understand the origin of the roll off. The aims of this work are thus quite important. And it is encouraging to see that the EQE remains high under high injection current density of 500 mA/cm². However, there are some assumptions and inconclusive arguments in the manuscript. More experiments need to be done to support the author's claims. Detailed issues of this manuscript are shown below:

Major issues:

1. It is very important to show the stability of the perovskite LEDs at high injection current density such as 500 mA/cm². In addition, the author should provide more voltage scanning details, such as scanning speed, dwell time at each voltage etc. These parameters may affect the EQE values. And, at what voltage does the efficiency start to roll-off and burn?
2. XRD measurements need to be done to confirm the phase of the film, and the orientation of features. For example, the PL looks quite heterogeneous (Fig. 3e) while EL (Fig. 4c) not.
3. This language of multiple quantum wells is unfounded, and inaccurate. In perovskite crystal systems, the structures that are being referred to are Ruddlesden-Popper phases. The transport properties of quantum wells are well known and described from work on III-V (mostly GaAs/AlGaAs) systems, but the authors make no attempt to prove that the films and structures being discussed follow these trends. It is very unlikely that they will. This language should be corrected, and removed, and in fact needs to be amended in the recent Nature Photonics paper as well that is cited so many times in this work. Further, if (see comment 2) the orientation shows the "quantum wells" to be perpendicular to the plane of the substrate, all the more problematic is this language.
4. When the molar ratio changes from 2:1:2 to 2:1.9:2, the composition in the solution changes from stoichiometric to non-stoichiometric with extra FAI. The extra organic ammonium halides will likely affect the optoelectronic property of the film and thus the device performance. The author should give an explanation on this. Also, it is important that the author can show morphology studies of both films, as it likely is changing, and may affect interpretation.
5. As for the field-induced charge separation, it is very speculative to claim that the charge separation follows a charge tunneling process. It is more likely that the carriers just transport parallel to the quantum walls, not tunneling through the walls (see comment 3)
6. The author should explain why the PLQE drops when the applied voltage increases from 0.75V to 1.5V. Is this repeatable? To base so much insight on a single data point is reckless. This so-called "trough" (an artifact of a b-spline) needs to be filled in with at least 4-5 points to be considered credible.
7. It is not convincing and very speculative that, in page 7, the EL emission at high forward bias is due to the screening effect. I do not think the bias and photoexcitation intensity dependent PLQE can support this argument. It is also possible that the photoexcited carriers fill the traps and PLQE increase at high excitation intensities.
8. Authors need to provide the absolute values of PLQE and EQE in Figure 1b. It is not convincing to make conclusion based on normalized values.

Minor issue:

1. Why most figures, such as Fig. 1b, Fig. 2a, have only few data points? In figure 4b, it is important to show all the data points.
2. I would suggest the author compare the PLQE of the LEDs at open circuit condition and at different voltages, particularly at 0.75V.
3. Intro: "3D perovskites suffer" should be "3D perovskites can suffer" as this is clearly a very limited citation, and is not an intrinsic point.

4. Is NMAI amine or ammonium? That's a pretty major difference.

Reviewer #2 (Remarks to the Author):

The authors present photo-luminescence (PL) and electro-luminescence (EL) measurements on perovskite light-emitting diodes (PeLEDs) in order to identify the physical mechanism causing the LED efficiency roll-off at high bias. Their interpretations and explanations are mainly based on the understanding and the terminology developed for GaN-based LEDs. However, this approach is inappropriate as outlined in the following:

- (1) The term quantum well (QW) refers to the formation of singular energy levels in 2-5nm thick InGaN/GaN layers. Here, it is incorrectly applied to a 30nm thick light-emitting perovskite layer without showing evidence of quantum levels.
- (2) PL measurements only make sense if the injected photons are absorbed only in the light-emitting perovskite layer, which is not justified for the 445nm laser source used here.
- (3) The phenomenological term "luminescence quenching" can have many different meanings and cannot serve as explanation for the LED efficiency roll-off, unless it is more clearly defined by the authors.
- (4) There is no evidence that Auger recombination really happens here, as claimed by the authors. Experimental evidence is needed by measuring high-energy carriers as in [10].
- (5) Previous investigations of the GaN-LED efficiency roll-off (droop) reveal that different mechanisms (carrier leakage, Auger recombination, defect recombination, free-carrier absorption) can lead to similar experimental results. The PL and EL measurements used here were shown to be insufficient for tracking down the actual mechanism. In other words, most of the explanations given here are highly speculative and not accompanied by convincing experimental evidence.

Reviewer #3 (Remarks to the Author):

This paper deals with quantum efficiency roll-off in perovskite, multiple quantum well, light emitting diodes. Authors use a combination of electroluminescence and photoluminescence measurements on an operational device, and conclude that the efficiency roll-off in this type of LED is caused by Auger recombination. Moreover, by increasing the width of the quantum wells, this effect can be suppressed. This leads to a perovskite LED with an external quantum efficiency of 12.7% with a minimal efficiency roll-off.

The paper is clear, follows a logical order and is well written. There is sufficient proof for the major claims of the paper, and I recommend publication. I hope the authors take into account the following comments:

In the introduction I miss a few sentences on possible technological advantages of perovskite LEDs.

There are other non-toxic LED technologies, which perform better, e.g. OLEDs.

The reduction in PL at negative bias is attributed to field induced charge separation. Have IV curves under illumination been measured? These would confirm (or disprove) the hypothesis of the authors, as the charges would flow out of the device as electrical (photo-)current.

"PL starts to increase after the bias exceeds the turn-on voltage (1.5 V). How is the "turn-on voltage" defined exactly? Figure 1 shows that current is already injected at 1V or even lower.

In a recent paper, the influence of electrical stress on perovskite LEDs is discussed (DOI: 10.1002/adma.201605317). In the current paper, I cannot find multiple scans of IV curves of the same device. Are there similar effects for this type of quantum well perovskite LEDs? If so, this should be discussed.

Point-by-Point Response to Referees

Reviewer #1:

Comment #1: In this paper, the authors fabricate perovskite LEDs with low efficiency roll-off at high injection current density, and attempt to understand the origin of the roll off. The aims of this work are thus quite important. And it is encouraging to see that the EQE remains high under high injection current density of 500 mA/cm². However, there are some assumptions and inconclusive arguments in the manuscript. More experiments need to be done to support the author's claims. Detailed issues of this manuscript are shown below.

Response: We thank the reviewer for the insightful comments and for recognising the importance of our work. Guided by these constructive comments, we have made many important improvements and clarifications throughout the paper.

Comment #2: It is very important to show the stability of the perovskite LEDs at high injection current density such as 500 mA/cm². In addition, the author should provide more voltage scanning details, such as scanning speed, dwell time at each voltage etc. These parameters may affect the EQE values. And, at what voltage does the efficiency start to roll-off and burn?

Response: We thank the referee for the helpful suggestions. To address this point, we have added the following results and discussion about stability under high electrical injection condition in the revised manuscript (Line 21 to 22, Page 11, and Line 1 to 3, Page 12, **Supplementary Fig. 7**, highlighted); "The device lifetime is dependent on the current densities, as shown in **Supplementary Fig. 7**. The lifetime (T_{50} , time to half of the initial brightness) of the 2:1.9:2 device (with a simple glass-epoxy encapsulation) under a constant current density of 100 mA cm⁻² (~3.0 V) is 30 min."

The scanning speed is 0.05 V/s and the dwell time is 1 s at each voltage step. As the reviewer suggested, we have added more voltage details about the measurements in

the revised manuscript. In **Figure 4a and 4b**, the peak EQEs of the 2:1:2 and 2:1.9:2 devices occur at 30 and 80 mA/cm² (~2.1 and ~2.3 V) respectively, and then they start to decrease. Considering the scanning speed and the EQE droop under less than 2.5 V, we believe that the degradation effect will not affect the measurement results. We have included these details in the revised text (Line 5 to 7, Page 11, and Line 1, Page 27, highlighted).

Supplementary Figure 7. Stability data for the 2:1.9:2 Perovskite MQW LEDs at different constant current densities.

Comment #3&4: XRD measurements need to be done to confirm the phase of the film, and the orientation of features. For example, the PL looks quite heterogeneous (Fig. 3e) while EL (Fig. 4c) not. This language of multiple quantum wells is unfounded, and inaccurate. In perovskite crystal systems, the structures that are being referred to are Ruddlesden-Popper phases. The transport properties of quantum wells are well known and described from work on III-V (mostly GaAs/AlGaAs) systems, but the authors make no attempt to prove that the films and structures being discussed follow these trends. It is very unlikely that they will. This language should be corrected, and removed, and in fact needs to be amended in the recent Nature Photonics paper as well that is cited so many times in this work. Further, if (see comment 3) the orientation shows the "quantum wells" to be perpendicular to the plane of the substrate, all the more problematic is this language.

Response: We appreciate the referee for the comments which have helped us to improve the paper. We agree that the quasi-two-dimensional perovskite structure can be referred to Ruddlesden-Popper phase. To our knowledge this usually refers to layered perovskite with a well-defined composition (*Acta Crystallogr.* **10**, 538–539 (1957), *Crystallogr. Rep.* **45**, 792–798 (2000)). Our perovskite is an assembly of quasi 2D perovskites with different bandgaps consisting of different layer number (n) of PbI_4 , which can be confirmed by optical measurement (Figure 1c, ref. 4). In the 2:1:2 film, the majority of the composition are layered perovskites with $n=2$, corresponding to a strong extinction absorption peak at 2.18 eV. In another words, our perovskite film is a mixture of layered perovskites rather a single phase perovskite. For this reason, we believe that Ruddlesden-Popper phase may not be the best description for the material system we report here.

Figure from ref. 4 (*Nat. Photonics* 10, 699–704 (2016)). Fig. 1c (Absorption and PL spectra of the MQW film. The PL spectrum is plotted in log scale (blue curve) and linear scale (red curve). The three pairs of the dashed lines (from left to right) correspond to the absorption (black) and PL (red) peaks from the QWs with $n=1$, 2 and 4, respectively)

Meanwhile, the layered perovskites are "quantum wells" when their energetic structures are considered. They are consisting of inorganic potential well layers and organic potential barrier layers, which are analogous to conventional inorganic semiconductor quantum wells. Their dielectric confinement effect and exciton binding energy have been reported in literature (*Solid State Commun.* **69**, 933–936 (1989); *Phys. Rev. B* **42**, 11099–11107 (1990)). Moreover, the term "quantum wells" was adapted from a wide range of papers concerning layered perovskites (*Solid State*

Commun. **84**, 657–661 (1992); *Phys. Rev. B* **47**, 2010–2018 (1993); *Phys. Rev. B* **51**, 14370–14378 (1995); *Sci. Technol. Adv. Mater.* **4**, 599–604 (2003); *Chem. Mater.* **28**, 2852–2867 (2016)).

Regarding the nano-morphology and the structure of the perovskite films, we also agree that the lack of clarity and XRD results in our previous manuscript makes some of our arguments not very well supported. To address this, we have included new results and discussions in the revised manuscript. Some of the details are provided below.

- 1) According to the reviewer's suggestion we have now included the XRD data of 2:1:2 and 2:1.9:2 perovskite films (**Supplementary Fig. 5**, highlighted) in the revised manuscript. The XRD spectra confirm that the perovskite films consist of small-*n* layered perovskites (11.0° , 14.5° , 16.2° , 25.7° and 29.3°) and large-*n* perovskites (13.9° and 28.1°). The peaks at 13.9° and 28.1° are consistent with the diffraction from (111) and (222) crystal planes of 3D FAPbI₃ (*Chem. Mater.* **26**, 1485–1491 (2014)). Moreover, the XRD measurement shows that the 2:1.9:2 perovskite film contains more large-*n* QWs with large grain size due to the more small cations incorporated in the precursor solution.

Supplementary Figure 5. XRD data of the 2:1:2 and 2:1.9:2 perovskite MQW films.

- 2) The high-resolution transmission electron microscopy (HRTEM, Figure 2 of ref. 4) shows that the majority of large-n perovskites are located at the top of the perovskite films. X-ray spectroscopy (EDX) elemental mapping measurement shows that the large-n perovskites and small-n perovskites locate at the interfaces between perovskite/TFB and perovskite/ZnO, respectively. The large-n perovskite region is much narrower than the small-n QW region, which presumably is the n=2 perovskites, and a gradually graded or mixed region is in between. The HRTEM measurement results also indicate that the layered perovskites are parallel to the plane of the substrate.

Figure from ref. 4 (Nat. Photonics 10, 699–704 (2016)). Fig. 2c (EDX mapping. Colour-mixed EDX mapping images (scale bar, 50 nm) present the element distribution of Pb, I and Zn. The normalized EDX count distribution of Pb and I across the perovskite layer are also presented.).

- 3) For PL measurement, the excitation wavelength is 445 nm which is mainly absorbed by the perovskites with n=2. Due to the energy transfer process (Figure 1d-g, ref. 4), the PL emission is mainly from large n perovskites. The weak PL emission from large bandgap perovskite suggests that the energy transfer process is not complete. The PL from different n perovskites can be more clearly presented in low temperature, as shown in the figure below. It clearly demonstrates that the perovskite film has a cascade energy structure formed by layered perovskites with different n values.

PL spectra of 2:1:2 NFPI₇ film at 6 K. The PL from perovskite structures with different n values shows multiple peaks under low temperature.

- 4) While in the EL measurement, electrons and holes are injected into the perovskite layer and accumulated at the large-n perovskites because of the cascade energy structure of the perovskite films (**Figure 1a**). In other words, the injected charges can directly recombine at the very thin layer of large-n perovskite. This is consistent with our finding that the weak emissions at higher energies observed in the PL spectrum are absent in the EL measurements.

Figure 1a. Schematic representation of the flat-band energy level diagram and structures of the 30 nm thick perovskite MQW film.

In light of the reviewer's comments, to clarify these points, we have revised our manuscript in the introduction. (Line 13 to 21, Page 3, highlighted). "The MQW perovskite film is an assembly of different layered perovskites which are "quantum wells" by considering their energetic structures. They are consisting of inorganic

potential wells and organic potential barriers. The bandgap of each quantum well is mainly determined by the number of PbI_4 monolayer sheets (n) within a layered perovskite. We note that the layered perovskite structure with a well-defined n number can be referred to Ruddlesden-Popper phase^{8,9}. High EQE up to 11.7% can be obtained with the MQW perovskites based LED because the lower bandgap regions which generate EL are effectively confined by higher energy gap perovskites⁴."

Comment #5: When the molar ratio changes from 2:1:2 to 2:1.9:2, the composition in the solution changes from stoichiometric to non-stoichiometric with extra FAI. The extra organic ammonium halides will likely affect the optoelectronic property of the film and thus the device performance. The author should give an explanation on this. Also, it is important that the author can show morphology studies of both films, as it likely is changing, and may affect interpretation.

Response: We thank the reviewer for raising this important point. As discussed previously, our perovskite film is a mixture of QWs consisting of layered perovskites with different n values. In other words, our MQW perovskite films are not formed by single-phase perovskite (with a fixed n value). When the composition in solution changes from 2:1:2 to 2:1.9:2, it will form more larger- n QWs in the perovskite films due to the larger quantity of small organic cations. This can be confirmed by the XRD measurement and the red-shifted PL spectrum (**Fig. 3e**), and also is consistent with the SEM measurements. Figures below show that the 2:1.9:2 film shows larger domain. To address the reviewer's comment experimentally, we have added the SEM and XRD data (**Supplementary Figs. 5 and 6**, highlighted) in the revised manuscript.

Supplementary Figure 6. SEM images of the Perovskite MQW films. a, 2:1:2. b, 2:1.9:2. Scale bar: 1 μm .

Comment #6: As for the field-induced charge separation, it is very speculative to claim that the charge separation follows a charge tunneling process. It is more likely that the carriers just transport parallel to the quantum wells, not tunneling through the walls (see comment 4).

Response: We thank the reviewer for this comment. Our "quantum wells" are parallel to the plane of the substrate, as discussed in our response to Comments #3 and #4 above. Besides, we performed PL decay measurements under different bias (Fig. 2c), which shows that the lifetime decreases as the electrical field strength increases. This suggests that the separated charges are removed from the QWs, otherwise the separated charges have the opportunity to radiatively recombine in the well, resulting in a longer PL decay lifetime. In addition, we have performed device JV scan under various illumination intensities. As shown below, under reverse bias the photocurrent increases with increasing illumination intensity, which clearly suggests a field induced charge separation and tunneling process.

Figure 2b. Dependence of current density on the driving voltage under various illumination intensities.

Comment #7: The author should explain why the PLQE drops when the applied voltage increases from 0.75 V to 1.5 V. Is this repeatable? To base so much insight on a single data point is reckless. This so-called "trough" (an artifact of a b-spline) needs to be filled in with at least 4-5 points to be considered credible.

Response: We thank the reviewer for raising these concerns. We should clarify that the curves in Figure 1b and similar figures are indeed plotted from many data points

(see the below figure). For labeling the curves with good clarity, only a few sparsely spaced markers (symbols) on the curves are displayed.

Left, Figure 1b of the manuscript, which presents curves with a few data points labeled by triangles, squares and circles. Right, the same curves with all data points labeled by shapes.

Under a forward bias of 0.75 V, the internal electric field across the perovskite film is minimized (approaching zero), leading to minimum luminescence quenching. When the bias is swept from 0.75 V to 1.5 V, the PLQE drops due to the increasing electrical field across the perovskite layer. We have now included these details in the text (Line 7 to 8, Page 6, highlighted). The relationship between PLQE and voltage is reproducible in different devices (see the below figure). We chose a representative figure to be included in the manuscript.

Dependence of current density, normalized PLOE and EOE on the driving voltage for different devices. They show similar trends.

Comment #8: It is not convincing and very speculative that, in page 7, the EL emission at high forward bias is due to the screening effect. I do not think the bias and photoexcitation intensity dependent PLQE can support this argument. It also possible

that the photoexcited carriers fill the traps and PLQE increase at high excitation intensities.

Response: We agree with the reviewer that the increased emission efficiency can be due to either screening effect or trap filling effect. However, in our MQW perovskites film, the trap filling effect is only important when the excitation intensity is very low (detailed discussion can be found in ref. 4), which can be confirmed by light-intensity-dependent PLQE measurement. As shown in **Fig. 3c**, when the excitation is below a low intensity of $\sim 0.3 \text{ mW cm}^{-2}$, the PLQE rapidly increases as the increasing excitation intensity due to the trap filling effect. Therefore, at high excitations (high voltages), we believe the enhanced emission efficiency is due to the screening effect.

Figure 3c. Excitation-intensity-dependent PLQEs of the 2:1:2 and 2:1.9:2 NFPI₇ MQW films under continuous-wave laser excitation.

Comment #9: Authors need to provide the absolute values of PLQE and EQE in Figure 1b. It is not convincing to make conclusion based on normalized values.

Response: We appreciate this suggestion. The efficiency roll-off in LEDs can be caused by either a reduction of PL efficiency or an excessive population of charge carriers passing through the device without forming electron-hole pairs. In this work we measure the change of PL intensity as the LED device is in operation. The PL is excited by using a chopped laser with very low intensity. The EL-PL signal was collected by using a silicon photodiode, and sent to a lock-in amplifier to distinguish the PL and EL signal. Although currently our system is not able to simultaneously measure the absolute values of PLQE and EQE, the fact that the decreasing PL

intensity at high bias tracks the EL EQE reveals that the EQE droop is due to the reduction of PL efficiency. Nevertheless, we measure EQE separately by using an integrating sphere, as we did in device characterization, and absolute values of film PLQEs, as shown in **Figure 3c** and **Figure 4b**.

Comment #10: Why most figures, such as Fig. 1b, Fig. 2a, have only few data points? In figure 4b, it is important to show all the data points.

Response: Again we appreciate the question about data points in figures. There are a large number of data points in these figures, and the curves in these figures are plotted based on all the data points, as shown below. We only used a few symbols to label the curves for better clarity. We value the reviewer's opinion but we think our current figures are able to show better details, especially when the curves are positioned closely.

Figure 2a. Left, Figure in the manuscript. Right, the same data set with all data points displayed.

Figure 4b. Left, Figure in the manuscript. Right, the same data set with all data points displayed.

Comment #11: I would suggest the author compare the PLQE of the LEDs at open circuit condition and at different voltages, particularly at 0.75V.

Response: As the reviewer suggested, we have compared the PL intensity at open circuit and at different voltages in the figure below. It shows that the PL intensity at open circuit is lower than that at ~ 0.75 V where we consider a flat band condition is achieved. The internal field quenches the PL at open circuit, while the PL loss is minimum under the flat band condition at ~ 0.75 V.

Dependence of current density, normalized PLQE and EQE on different voltages.

Comment #12: Intro: "3D perovskites suffer" should be "3D perovskites can suffer" as this is clearly a very limited citation, and is not an intrinsic point.

Response: We thank the reviewer for the useful suggestion. We have changed the description in the revised text (Line 10, Page 3, highlighted) accordingly.

Comment #13: Is NMAI amine or ammonium? That's a pretty major difference.

Response: NMAI is ammonium. To clarify this issue, we have provided the full name of NMAI in the text (Line 13, Page 5, highlighted).

Reviewer #2 (Remarks to the Author):

Comment #1: The authors present photo-luminescence (PL) and electro-luminescence (EL) measurements on perovskite light-emitting diodes (PeLEDs) in order to identify the physical mechanism causing the LED efficiency roll-off at high bias. Their interpretations and explanations are mainly based on the understanding and the terminology developed for GaN-based LEDs. However, this approach is inappropriate as outlined in the following.

Response: We thank the reviewer for the critical comments which enabled us to improve the quality and clarity of our revised manuscript. We agree that our perovskite-based LEDs share some similarities with GaN-based LEDs. To our knowledge, the MQW GaN-based LEDs require precise control over lattice parameters matching to avoid electronic defects that often cause trapping and quenching of the excited states. Our MQW perovskites are self-organized nanostructures prepared from low-temperature solution processing. They exhibit a low level of electronic defects comparable to that of single-gap single-crystalline semiconductors produced using advanced processing techniques. The efficiency roll-off of perovskite LEDs at high emission intensities can be caused by several reasons as in GaN-LED. In general, the EQE droop under high bias can be caused either by leakage current or a reduction of luminescence efficiency (luminescence quenching). In order to separate these effects in MQW perovskite LEDs, we simultaneously measure EL and PL on a working LED device. We find that under high bias voltage, the EL EQE tracks the PL intensity, which suggests that the efficiency roll-off in PeLEDs is mainly due to the reduction of luminescence efficiency. Further evidenced by time-resolved PL measurement, we attribute the luminescence quenching to the nonradiative Auger recombination. Moreover, we demonstrate the Auger recombination can be suppressed by increasing the width of perovskite quantum wells, resulting in record high efficiency perovskite LEDs with EQEs of up to 12.7% and significantly reduced roll-off. We hope our clarification and

the improvements we made in the paper would help us to communicate our results to the reviewers and the readers.

Comment #2: The term quantum well (QW) refers to the formation of singular energy levels in 2-5 nm thick InGaN/GaN layers. Here, it is incorrectly applied to a 30 nm thick light-emitting perovskite layer without showing evidence of quantum levels.

Response: Thanks to the reviewer's comment, we have realised that our usage of the terminology was not well explained in the previous version of our manuscript, and we have expanded the introduction section to provide more details about our MQW structure (Line 13 to 21 Page 3, highlighted). The 30 nm perovskite is a mixture of QWs consisting of layered perovskites with different n values. The layered perovskites are "quantum wells" by considering their energetic structures. They are consisting of inorganic potential wells and organic potential barriers, which are analogous to conventional inorganic semiconductor quantum wells. Its dielectric confinement effect and exciton binding energy have been well studied in literature (*Solid State Commun.* **69**, 933–936 (1989); *Phys. Rev. B* **42**, 11099–11107 (1990)). Moreover, the term "quantum wells" have been widely used in literature on layered perovskites (*Solid State Commun.* **84**, 657–661 (1992); *Phys. Rev. B* **47**, 2010–2018 (1993); *Phys. Rev. B* **51**, 14370–14378 (1995); *Sci. Technol. Adv. Mater.* **4**, 599–604 (2003); *Chem. Mater.* **28**, 2852–2867 (2016)). In addition, the optical and XRD measurements show that our perovskite is an assembly of quasi 2D perovskites with different bandgaps consisting of different number (n) of layers of PbI_4 . The high-resolution transmission electron microscopy (HRTEM) indicates that the layered perovskites are parallel to the plane of the substrate with the majority of large- n perovskites locating at the top of the perovskite films. The PL measurement further confirms that the perovskite film has a cascade energy structure from different- n of layered perovskites. The EL measurement shows the injected charges can directly recombine at the very thin layer of large- n perovskite. A detailed description of the MQW structure of our perovskite films can also be found in our previous work (*Nat. Photonics* **10**, 699–704 (2016)).

Comment #3: PL measurements only make sense if the injected photons are absorbed only in the light-emitting perovskite layer, which is not justified for the 445 nm laser source used here.

Response: To address this important point raised by the reviewer, we present our absorption data below. The cut-off absorptions of TFB and ZnO layers in our device are ~430 and ~375 nm, respectively (see the below figure). So the 445 nm excitation light can only be absorbed by the light-emitting perovskite layer.

Absorption spectra of ZnO and TFB.

Comment #4: The phenomenological term "luminescence quenching" can have many different meanings and cannot serve as explanation for the LED efficiency roll-off, unless it is more clearly defined by the authors.

Response: We thank the reviewer for this comment. Here the "luminescence quenching" means a reduction in luminescence efficiency due to non-radiative recombination or charge separation. We have clarified this in the revised manuscript (Line 13 to 14, page 4, highlighted).

Comment #5: There is no evidence that Auger recombination really happens here, as claimed by the authors. Experimental evidence is needed by measuring high-energy carriers as in [10].

Response: We thank the reviewer for this valuable comment. In principle, the direct measurement of the generation of Auger electrons would be possible as shown in ref. 10 (ref. 17 in the revised manuscript). However, our concern was the suitability of this

method to the system we investigated here. In our LED device, the full active area is covered by Au electrode. The Auger electrons created in the emission layer have to travel through the metal electrode, and it will be difficult to be measured because of the scattering in the metal layer. Moreover, even in ref. 10, this method is subject to controversial discussion (arXiv:cond-mat/1305.2512).

On the other hand, Auger recombination has been well recognized in perovskites (*Acc. Chem. Res.* **49**, 146–154 (2016), *Adv. Mater.* **26**, 1584–1589 (2014); *Adv. Mater.* **27**, 7938–7944 (2015)). In the current manuscript, we show that under high bias voltage, the EL EQE tracks the PLQE, and the PLQE and PL lifetime simultaneously decrease at high excitation intensities, which is consistent with Auger recombination in high excitations, as suggested in the above literature.

Comment #6: Previous investigations of the GaN-LED efficiency roll-off (droop) reveal that different mechanisms (carrier leakage, Auger recombination, defect recombination, free-carrier absorption) can lead to similar experimental results. The PL and EL measurements used here were shown to be insufficient for tracking down the actual mechanism. In other words, most of the explanations given here are highly speculative and not accompanied by convincing experimental evidence.

Response: We appreciate this comment and we agree with the referee that there are a number of possible efficiency roll-off mechanisms beside the Auger mechanism, which will compete with each other in the device. However, we can assess the importance of these mechanisms in working devices based on our experimental results. For carrier leakage, it will not cause PLQE to track the EQE reduction at higher voltages. For defect recombination, we have shown that the defect density is low in MQWs (*Nat. Photonics* **10**, 699–704 (2016)) and the excitation-intensity-dependent PLQE in **Fig. 3c** shows that the traps were filled at very low excitation intensities. Therefore, the defect recombination is only important when the current density is low. Free-carrier or excited-state absorption spectrum has a peak at 750 nm in our MQW perovskites, as we shown earlier (ref. 4, *Nat. Photonics*

10, 699–704 (2016)), while the EL emission peaked at 786 nm. If the free-carrier absorption is significant at high excitations, then we would expect the EL emission peak will be red-shifted due to the more absorption in short-wavelength part. However, we did not observe this, and the EL spectra are almost identical under various bias (see the below figure). Therefore, it is reasonable to attribute the main reason for the EQE droop in MQW perovskite LEDs to Auger recombination.

a, Photo-induced changes in transient absorption spectra of NFPI₇ film (Fig. 1f, *Nat. Photonics* **10**, 699–704 (2016)). *b*, EL spectra of 2:1:2 NFPI₇ device under various voltages.

Reviewer #3 (Remarks to the Author):

Comment #1: This paper deals with quantum efficiency roll-off in perovskite, multiple quantum well, light emitting diodes. Authors use a combination of electroluminescence and photoluminescence measurements on an operational device, and conclude that the efficiency roll-off in this type of LED is caused by Auger recombination. Moreover, by increasing the width of the quantum wells, this effect can be suppressed. This leads to a perovskite LED with an external quantum efficiency of 12.7% with a minimal efficiency roll-off.

The paper is clear, follows a logical order and is well written. There is sufficient proof for the major claims of the paper, and I recommend publication. I hope the authors take into account the following comments.

Response: We thank the reviewer for the supportive comments, which have encouraged us to improve our paper.

Comment #2: In the introduction I miss a few sentences on possible technological advantages of perovskite LEDs. There are other non-toxic LED technologies, which perform better, e.g. OLEDs.

Response: In light of the review comment, we have considered the main advantages of perovskite LEDs. Compared to other similar technologies, such as organic LEDs and quantum dot LEDs, perovskite LEDs can potentially achieve better efficiency at high brightness since its charge mobility is higher and luminescence quenching at high exciton concentration is less significant. We have added this in the introduction (Line 5 to 8, page 3, highlighted).

Comment #3: The reduction in PL at negative bias is attributed to field induced charge separation. Have IV curves under illumination been measured? These would confirm (or disprove) the hypothesis of the authors, as the charges would flow out of the device as electrical (photo-) current.

Response: We thank the reviewer for this critical comment. We have added the J-V curves of the device under various illumination intensities in the revised text (**Fig. 2b**). It clearly shows that the charges can separate at negative bias under illumination (Line 3 to 5, Page 7, highlighted).

Figure 2b. Dependence of current density on the driving voltage under various illumination intensities.

Comment #4: "PL starts to increase after the bias exceeds the turn-on voltage (1.5 V). How is the "turn-on voltage" defined exactly? Figure 1 shows that current is already injected at 1 V or even lower.

Response: We thank the reviewer for this comment. Unlike the turn-on voltage for conventional p-n diodes, the turn-on voltage of the LEDs is the onset of the radiance-voltage curve. It's the voltage when light emission from device can be measured. We have added the definition in the text (Line 18, Page 7, highlighted).

Comment #5: In a recent paper, the influence of electrical stress on perovskite LEDs is discussed (DOI: 10.1002/adma.201605317). In the current paper, I cannot find multiple scans of IV curves of the same device. Are there similar effects for this type of quantum well perovskite LEDs? If so, this should be discussed.

Response: We appreciate the reviewer for raising this interesting point. As shown in the figure below, after 14 electrical scans, there is no significant change of the J-V curves with the MQW PeLEDs. In the MQW perovskites, the ion migration is not as serious as the 3D perovskite due to the existence of the large organic cations (*ACS Energy Lett.* **2**, 1571–1572 (2017)).

Current density versus voltage curves of a MQW PeLED device measured with continuous electrical scans.

REVIEWERS' COMMENTS:

Reviewer #1 (Remarks to the Author):

After re-reading the manuscript and reading the response to reviewers, I simply remain unconvinced about (1) the quantum well language, and (2) the claim that Auger recombination is at fault for rolloff. I find the PLQE measurement of the functioning LED to be very interesting, perhaps might even say fascinating, but find the explanations unconvincing to say the least. This is a tough experiment, and demands more thorough understanding to truly show the community something new. Right now, there are new data, but the explanations don't seem correct.

Reviewer #2 (Remarks to the Author):

Thank you for your detailed response to my concerns. I am satisfied with your answers and I support the publication of this article.

Reviewer #3 (Remarks to the Author):

The authors have adequately responded to most of my comments, and have put a lot of effort into collecting new data to support their hypothesis. This has made the manuscript certainly much stronger. For me, the paper can be published. I am however still not convinced by the definition of turn on voltage since the definition the authors use depends on the accuracy of the measurement setup used. If you have a better setup, you can measure lower light intensity emission, and the turn-on voltage would be measured to be lower with a sensitive setup than with a less sensitive setup.

Point-by-Point Response to Referees

Reviewer #1:

Comment: After re-reading the manuscript and reading the response to reviewers, I simply remain unconvinced about (1) the quantum well language, and (2) the claim that Auger recombination is at fault for roll off. I find the PLQE measurement of the functioning LED to be very interesting, perhaps might even say fascinating, but find the explanations unconvincing to say the least. This is a tough experiment, and demands more thorough understanding to truly show the community something new. Right now, there are new data, but the explanations don't seem correct.

Response: We appreciate the further comments from Reviewer 1. Although we remain confident about the interpretations of our results, we take the reviewer's view as an important guide to improving scientific rigor and clarity. To reflect this, we have made further clarifications of the terminologies used in the introduction section, stating the key difference between our solution-processed perovskite and conventional multiple quantum well materials. The term "multiple-quantum well (MQW)" is now removed from the abstract. In addition, we have now stated in the revised manuscript that the Auger recombination is one of the most likely reasons for roll-off in the revised manuscript, without excluding the possibility of other processes.

Reviewer #2:

Comment: Thank you for your detailed response to my concerns. I am satisfied with your answers and I support the publication of this article.

Response: We thank the reviewer for appreciating our work and for supporting the publication of our paper.

Reviewer #3:

Comment: The authors have adequately responded to most of my comments, and have put a lot of effort into collecting new data to support their hypothesis. This has made the manuscript certainly much stronger. For me, the paper can be published. I am however still not convinced by the definition of turn on voltage since the

definition the authors use depends on the accuracy of the measurement setup used. If you have a better setup, you can measure lower light intensity emission, and the turn-on voltage would be measured to be lower with a sensitive setup than with a less sensitive setup.

Response: We thank the reviewer for the supportive comments. We agree that the definition of turn-on voltage depends on the accuracy of the measurement setup. In our experiments, the increase of radiance is rather quick after device turn-on, and a voltage step of 0.05V corresponds to ~2 orders of magnitude increment of radiance (as shown in Fig. 4a). We believe the measurement error between different detectors shall be very small. However, we appreciate the reviewer's suggestion and have added the following statement in the manuscript. "We note that the observation of a slightly lower turn-on voltage may be possible with more sensitive experimental setups."